# Energy-Investment Decision-Making for Industry: Quantitative and Qualitative Risks Integrated Analysis

Eva M. Urbano *, Victor Martinez-Viol, Konstantinos Kampouropoulos and Luis Romeral

MCIA Research Center, Department of Electronic Engineering, Universitat Politècnica de Catalunya, Rambla de Sant Nebridi 22, 08222 Terrassa, Spain; victor.martinez.viol@upc.edu (V.M.-V.); konstantinos.kampouropoulos@upc.edu (K.K.); luis.romeral@upc.edu (L.R.)
* Correspondence: eva.maria.urbano@upc.edu

**Abstract:** Industrial SMEs may take the decision to invest in energy efficient equipment to reduce energy costs by replacing or upgrading their obsolete equipment or due to external socio-political and legislative pressures. When upgrading their energy equipment, it may be beneficial to consider the adoption of new energy strategies rising from the ongoing energy transition to support green transformation and decarbonisation. To face this energy-investment decision-making problem, a set of different economic and environmental criteria have to be evaluated together with their associated risks. Although energy-investment problems have been treated in the literature, the incorporation of both quantitative and qualitative risks for decision-making in SMEs has not been studied yet. In this paper, this research gap is addressed, creating a framework that considers non-risk criteria and quantitative and qualitative risks into energy-investment decision-making problems. Both types of risks are evaluated according to their probability and impact on the company's objectives and, additionally for qualitative risks, a fuzzy inference system is employed to account for judgmental subjectivity. All the criteria are incorporated into a single cost–benefit analysis function, which is optimised along the energy assets' lifetime to reach the best long-term energy investment decisions. The proposed methodology is applied to a specific industrial SME as a case study, showing the benefits of considering these risks in the decision-making problem. Nonetheless, the methodology is expandable with minor changes to other entities facing the challenge to invest in energy equipment or, as well, other tangible assets.

**Keywords:** decision-making; risk assessment; uncertainty; optimal energy design; prosumer

## 1. Introduction

The selection and management of assets are crucial for the achievement of enterprises' objectives in the industrial sector. Among the company's tangible assets, those related to energy generation and management have special interest due to their impact on production costs and thermal comfort. Currently, small-and-medium enterprises (SMEs), and particularly those in the manufacturing sector, have a high environmental footprint, and literature estimates that they contribute 60–70% of industrial pollution in Europe [1]. Therefore, equipment investment and operation of the SMEs are critical for the green transformation and can increase their growth performance [2]. However, the inclusion of new energy assets such as renewable energy sources (RES) and other supporting equipment to improve the competitiveness of enterprises and reduce the environmental footprint has not been studied adequately [3], and industries, especially SMEs, are facing difficulties in incorporating them in their energy infrastructure [4]. Besides, the energy transition that is already taking place presents an opportunity for the industrial sector to adopt an active role in transforming the energy market, for example, becoming a prosumer. This active role implies the establishment of a smart energy management strategy that would make us the industrial energy assets to meet internal demand while adapting their operation to external market conditions, generating a profit from this interaction and opening new

business models in the industrial entity. To be able to incorporate these strategies, it could be necessary, among other solutions, to perform an investment for upgrading the energy equipment and infrastructure of the industry through its re-design and sizing to use it as a productive asset. Due to their limited financial capacity and managerial system, industrial SMEs investments occur in discrete points in time, not prolonging the investment in multiple phases as performed by other entities such as governmental organizations or large companies, which can modify the project according to the evolution of industrial, legal or social boundary conditions [5]. Instead, SMEs' decisions are taken based on immediate investment return and maximization of profit along the lifetime of the equipment [6]. Therefore, industrial SMEs face the investment decision-making problem only with the current information and accepting the uncertainty related to the real situation evolution at which the upgraded infrastructure would operate. Moreover, some of the factors that are commonly employed as criteria in the decision-making process are hard to measure, and its mere definition presents levels of venture and hazard, as, for example, social acceptance and legislation alignment. Thus, the required investment for industrial SMEs to upgrade their energy infrastructure is inherently linked to risks arising from both the uncertainty in the future situation, which can be represented as a quantitative risk, and the measurement or subjectivity of some of the possible decision criteria, reflected as a qualitative risk. To support industrial SMEs in performing these investments, the research objective of this paper is to create a framework that addresses risk-informed decision-making (RIDM) for their energy investment problem. The specific research questions that have to be answered and that are addressed here are:

- Which risks and factors have to be treated for the energy investment RIDM problem in industrial SMEs and how can they be processed?
- Which methodology is suitable to address this RIDM problem?
- Which techniques and tools are convenient and how should be used for optimising the energy investment RIDM problem in industrial SMEs considering the previously addressed risks and factors?

To created framework to answer these questions, in the following paragraphs a review of the state-of-the-art on methodologies and techniques applied to RIDM processes and energy investment decisions is exposed.

Up to date, some RIDM approaches for general industrial applications have been presented in the literature. In [7], a methodology for decision-making considering quantitative and qualitative risk factors is presented with a focus on enterprises with serious health and environmental risk aspects such as mining, nuclear and aerospace industries. In this work, a set of alternatives exist, and the decision is taken by deliberation. In [8], a multi-criteria decision analysis (MCDA) is presented for planning the energy generation network of a country, selecting the best option among the alternatives employing an analytical hierarchy process (AHP). Although a cost–benefit analysis (CBA) would have been suitable for this case, it is argued that qualitative attributes are difficult to transform and incorporate in the final functions.

In CBA, advantages and disadvantages accounting for different criteria over the lifetime of investment alternatives are both assessed and incorporated in a single function [9], which can be optimised to reach either the best value of the investment or the best benefit to cost ratio. In [10], both quantitative and qualitative parameters are included in the CBA, although qualitative attributes are set as crisp numerical values without considering the vagueness of qualitative judgements. Additionally, the weight selection methodology is not clear, stating that the application of weights to compare qualitative and quantitative data is difficult and presents a barrier to the development of CBAs. This weighting issue is solved in [11], where an AHP is employed to weight the criteria and ease the selection of the best alternative through an MCDA. AHP enables to structure the decision-making problem according to a hierarchy of preferences from which each of the weight of the criteria, which can be of various natures and have different units, is obtained through the analysis performed by decision-makers [12]. All these RIDM problems presented until

now deal with a discrete number of alternatives, and qualitative and quantitative risks considered are transformed to crisp and precise values. However, qualitative measurements are subject to judgmental vagueness and thus their consideration as crisp numbers cause loss of information. In the past, an alternative to deal with qualitative values' vagueness for decision-making was presented based on a fuzzy approach, which transformed the linguistic risk appreciations into continuous numerical functions [13]. In this work, however, only qualitative fuzzy parameters were employed to assess the risk of construction projects, omitting quantitative information, which is by its own nature much more precise. Although the exposed RIDM approaches have addressed the investment problem for some industrial applications, a suitable framework for industrial SMEs' RIDM energy investment and optimisation problem has not been developed yet.

In the general field of energy investment including industrial, services and residential sectors, research has been performed focusing on energy design and planning without analysing the associated risks [14,15]. Although in some cases the uncertainty of the output is studied after performing the decision, RIDM is not carried out. This is the case of [16], where the performance of a hybrid energy system is analysed under uncertain events; and of [17], where the response of an energy system is studied according to fluctuations in system inputs', such as the cost of energy. The literature on energy investments considering risks inside the decision-making problem is scarce, although risk analysis is a common tool for companies. In [18], a life cycle cost (LCC) analysis is performed for a building energy system considering the risk related to economic parameters through Monte Carlo simulation. In [19], the design is done evaluating through the same technique the risk related to quantitative costs and technological aspects, and in [20], energy carriers price and investment costs uncertainties are considered. In all of these works, the risk is expressed employing a quantitative probability approach, focusing on economic parameters. However, real-world industries' decision-making problems include a mixture of criteria that are not easily quantifiable and have to deal with insufficient information, such as the contribution of the investment into social benefit or the future continuity of the enterprise, which makes it not possible to employ probabilistic methods [21]. This fact enhances the application of both quantitative and qualitative risk assessment techniques which have not been employed in the energy investment literature until now. Due to the investment characteristics and the inclusive growth role the SMEs play in society, as well as the requirements of energy assets to fulfil internal enterprise requirements over time and the possible adoption of an active energy role to open a new business models, it is required to create a methodology in which risks are correctly considered.

In this paper, a methodology to properly address the RIDM energy equipment investment problem considering the mixture of criteria that exist for industrial SMEs is proposed with the aim of improving their competitiveness and allow them to play an active role in the energy market. In this new methodology, both quantitative and qualitative risk must be assessed accounting for the judgemental vagueness of the decision-maker, while addressing the optimisation problem continuously over the operation time and space of possible combined solutions of the equipment to use rather than analyse only a few subjectively chosen alternatives. As a basis for solving this problem, a CBA approach is employed, which is suitable for the application in enterprise assets management problems [22]. In order to deal with real-world situations where a mixture of criteria exists, the proposed CBA approach incorporates both quantitative and qualitative data, being the latter assessed through a fuzzy approach to account for judgemental vagueness. These risks, together with the non-risk criteria to make the decision, are unified into a single objective function employing an AHP weighting technique that represents a balanced trade-off of the different factors considered in the RIDM energy investment problem. The objective function is evaluated and optimised continuously over time and also over the continuous space of solutions, analysing all the alternatives and not relying on a pre-specification of them. This procedure enables us to reach an optimal decision considering the specifications and constraints provided by the industrial SME.

Bearing in mind the state-of-the-art in RIDM and its application to the problem of energy investment for industrial SMEs, this paper presents the following main novelties and improvements:

- Creation of a methodology to support industrial SMEs in the energy investment decision-making process considering relevant factors and criteria to improve their competitiveness and accounting for related risks that could affect their performance.
- Optimisation of the RIDM energy investment problem including equipment options and its operation to address internal demand and produce a profit from exchanges with the energy market. To do this, the continuous-time operation of the SME and all possible combinations and sizes of energy equipment are evaluated.
- Evaluation of both qualitative and quantitative risks for energy-investment decision-making in a unique function to account for uncertain deployment scenarios and face the difficulty in the measurement of subjective criteria.

These novelties imply the adoption and usage of strategies, techniques and tools which have not been employed until now in RIDM for energy investment. These are considered as collateral paper contributions consequence of the previously stated ones, and are:

- Transformation of subjective criteria represented as qualitative risks into fuzzy sets to account for judgmental vagueness of industrial SMEs' decision-makers.
- Incorporation of qualitative and quantitative measurements into a single function expressed as CBA through AHP weighting, properly reflecting the preferences of decision-makers.

This paper is structured as follows. First of all, in Section 2, the proposed methodology for energy-investment decision-making is further explained. Secondly, in Section 3, a case study based on a real manufacturing industrial plant at which this methodology is applied is explored. The results of this case study and their discussion are shown in Section 4, and, lastly, conclusions are drawn in Section 5.

## 2. Energy-Investment Decision-Making Methodology

In this section, the methodology to assess the RIDM for industrial SMEs aiming to invest in energy assets to upgrade their energy infrastructure and improve their competitiveness is presented. Industrial SMEs are characterised by performing investments in discrete points in time to maintain or increase the productivity of their plant. For the case of investment in energy assets, their selection influences the long-term continuity of the enterprise as it affects the efficiency at which the production load is met as well as its impact on local social welfare and corporate image. However, the information with which industrial SMEs manage to perform these decisions present uncertainty both in the forecast of the future situation and in the measurement of qualitative decision-making criteria. These facts, together with decision-making difficulties involving access to financial sources, are challenges faced by SMEs worldwide [23]. Therefore, the proposed methodology to address and support SMEs' RIDM in energy investment, which can be seen in Figure 1, has been defined to be expandable to SMEs around the globe.

To implement this methodology, information regarding a specific industrial SME framework and the variables and constraints that apply are required. On the one hand, the specific SME internal and context information include:

- Production and energy consumption profiles;
- Local energy and emissions costs and applicable legislation;
- Available energy solutions and technological maturity of the company for using them; and
- Opinion and views of the local community on innovative energy infrastructures and equipment for renewable energy, who may have, for instance, different acceptance of photovoltaic and biomass due to their different landscaping and logistic impacts.

On the other hand, the constraints that apply for the energy-investment problem in industrial SMEs and that should be considered are:

- Limited initial investment;
- Required payback period;
- Geographic constraints; and
- Legislation constraints.

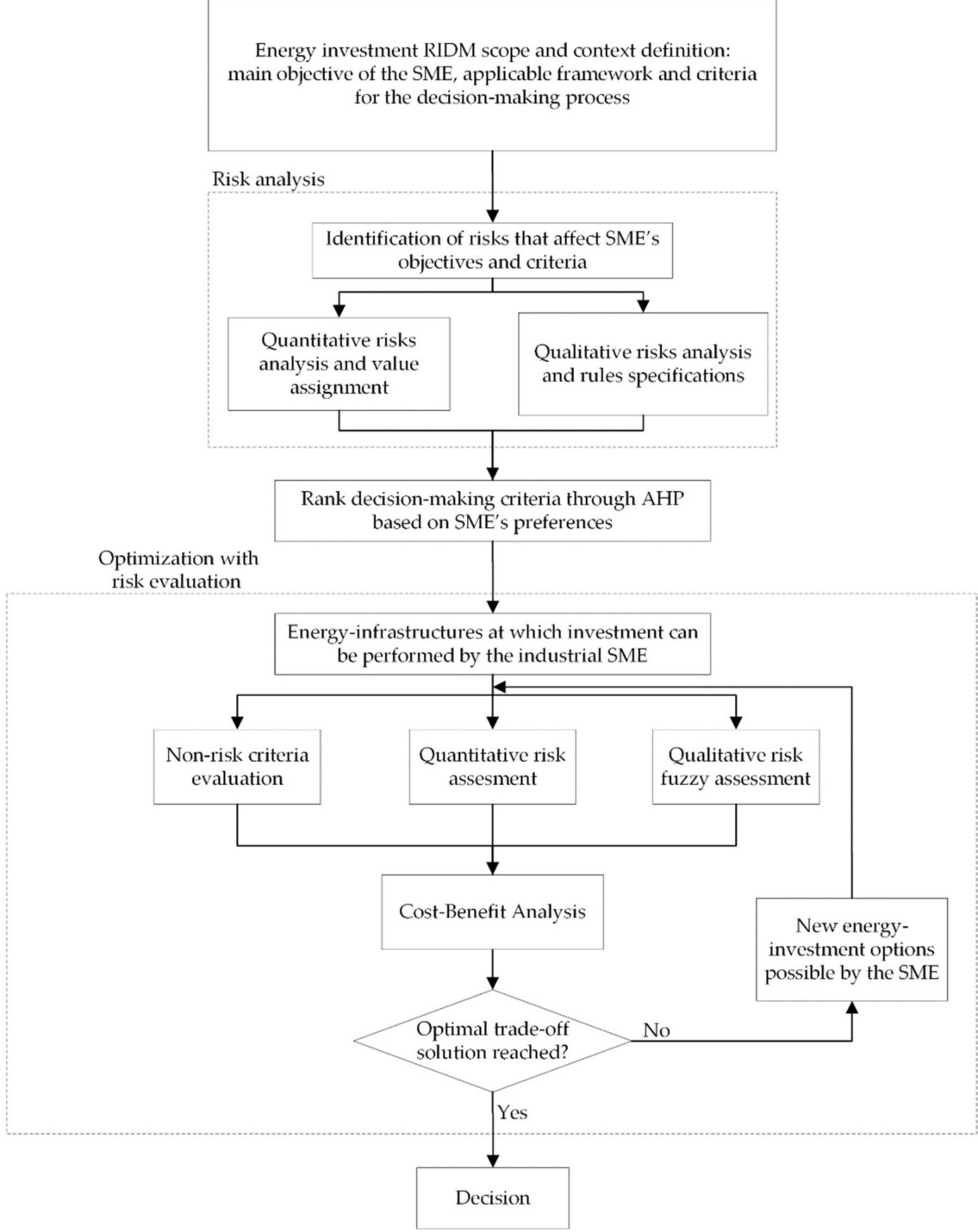

**Figure 1.** Energy-investment decision-making methodology.

These parameters and variables must be locally analysed and stated to process them according to their uncertain nature. Then, they serve as input for the optimisation problem, where the potential energy infrastructures in which the SME can invest are analysed, evaluating the identified risks and criteria. This evaluation of possible energy infrastructures is performed through an iterative algorithm, which analyses the output for each of them and moves towards the solution most suited to the studied industrial SME, the priorities of which are specified by decision-makers and adequately incorporated in the optimisation problem to reach the best trade-off solution.

In the following sections, each of the stages of the proposed methodology are exposed together with the techniques employed and their background.

### 2.1. Scope, Context and Criteria

Industrial SMEs face the problem of investing equipment to upgrade their energy infrastructure and include RES due to the required replacement of outdated energy assets or the existence of a socio-political framework that forces or encourages them to do so. The current economic, environmental and technical context is also opening the path and promoting the inclusion of distributed energy resources and active energy actors to achieve a cleaner and sustainable energy system [24]. For industries, it is possible to be part of this change by adopting a prosumer role, one way to do so being the upgrading of their energy infrastructure. However, the uncertain future market situation and the energy price volatility supposes a financial risk that inhibits industries to perform these investments [25]. For this reason, the methodology presented in this paper considers the relevant criteria to take into account for choosing the most suitable energy-investment solution, and the risks related to them.

The criteria represent the decision drivers to evaluate the potential energy-investment solutions. These criteria can be related to risks or not, being possible the following situations, or a combination of them:

- Non-risk criteria: Their value is computed objectively and it is not influenced by the uncertainty in the inputs of the system. In the RIDM energy investment optimisation problem, these non-risk criteria are selected according to the scope of the problem and can be, for example, the total emissions of the system if the emission factor is considered constant, which is a common approach in energy-investment optimisation problems [26].
- Criteria affected by inputs' uncertainty: The value of the criteria depends on uncertain inputs. These uncertainties have to be identified as quantitative risks, and the variation of the affected criteria according to them have to be computed. This variation is then included in the decision-making problem as an additional criterion aiming for its reduction, minimizing the risk at which the enterprise is exposed. For the case of energy investment problems in industrial SME, a common decision-making criterion is the net present value (NPV). The value of the NPV in the proposed energy infrastructure is influenced, among others, by the cost of energy carriers. As there is uncertainty in future energy costs that can be quantifiable, the variation of the NPV should be computed according to them and introduced in the optimisation problem.
- Subjective criteria: These criteria are difficult to assess mathematically, as they rely on subjective opinions and, consequently, their evaluation represents a risk by itself. To include them in the decision-making process, they are treated as qualitative risks employing a fuzzy methodology to account for judgemental vagueness. This is the case of criteria such as social acceptance, whose value relies on the knowledge about the local community where the SME is placed and the opinion based on the experience of decision-makers.

In this paper, and to properly address the mixture of criteria and risks present in the energy investment RIDM problem of industrial SMEs, the combination of criteria with both quantitative and qualitative risks is considered. For this problem, the non-risk criteria are related to factors arising from the operation of the upgraded energy infrastructure and

its economic and environmental impact, such as the obtained profit and emissions. To compute these parameters, the industrial plant is modelled mathematically and its operation optimised. Additionally, quantitative and qualitative risks related to the upgraded energy plant are evaluated following the indications of relevant research performed in the literature to date, including the fuzzy treatment of qualitative measurements.

The criteria and risks to decide the best trade-off energy-investment solution are selected according to specific enterprise interests and should include economic, environmental, technical and social aspects. A review of the criteria for energy investment evaluations commonly employed in the literature is available at [27], which can be modified and adapted to the specific problem treated. The scope of the energy-investment decision problem also has to be settled by the company, specifying the equipment considered for installation, the available space for installation and other limitations, the required risk detail, and any restrictions that apply, such as maximum initial investment, payback time, etc.

### *2.2. Risks Analysis*

Once the SME decides the criteria which are relevant for consideration in the energy-investment problem, the risks that affect them have to be identified. In this section, the methodology to classify and treat these risks is assessed.

### 2.2.1. Identification

The first stage in the risk analysis process is the identification of the risks present in the energy investment decision-making problem. Industrial SMEs are characterised by a management system where the owner of the enterprise acts, most of the time, as manager of the company, and there is a lack of a management body with suitable specialised knowledge for decision-making [6]. To successfully implement an energy-investment decision-making process, it is required to establish a decision-board either internally in the enterprise or resorting to external advisors. Once decision-makers have been established, the risk detection process has to be performed aiming to identify as many risks as possible according to the scope of the problem. The possibility of not identifying a risk due to a lack of knowledge or awareness is not assessed in this paper.

As mentioned in the previous section, risks can be embedded in the criteria or can be the effect of quantitative inputs uncertainty in the criteria. To properly deal with them, their probability and impact on the enterprise's objectives and criteria have to be addressed, reaching a risk evaluation measure [28]. The probability of a risk is the measure of how possible it is for an uncertain event to happen, and the impact refers to the effect that this event would cause on the performance of the energy infrastructure and the SME's objectives. In the following subsections, the definition strategy for both types of risks is exposed.

### 2.2.2. Quantitative Risks Definition

The steps to treat these risks in the decision-making process are exposed in Figure 2.

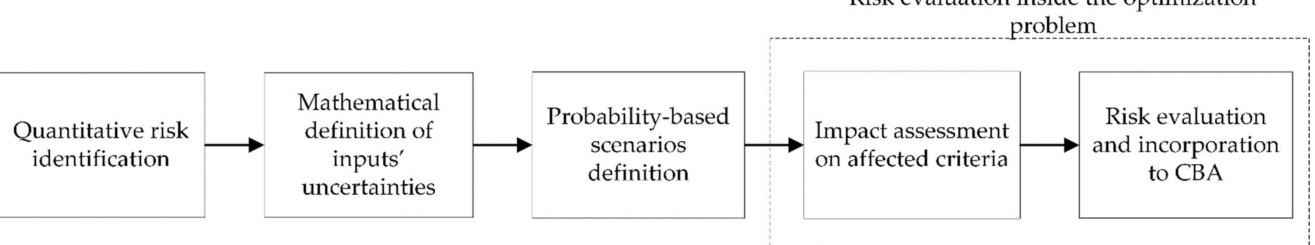

**Figure 2.** Quantitative risks treatment.

In the energy-investment decision-making problem for industrial SMEs, quantitative risks deal with the uncertainty related to the future energy situation, and include, among

others, future energy carrier and emissions costs. Once the decision-board identifies all the applicable risks for the specific problem considered, the inputs' uncertainties have to be expressed mathematically. The possible values that the uncertain inputs can take can be denoted as a set of discrete values with their corresponding probabilities [16], such as in the case of existent forecasting scenarios of future energy costs, or as continuous probability distribution functions [19] if a more detailed analysis is available. The type of expression depends on the nature of the risk and the information gathered. If a continuous probability distribution function is employed, this has to be transformed into a set of probability-based scenarios to be able to evaluate their impact on the criteria. This is done through the Monte Carlo sampling strategy, which is widely used and accepted in RIDM processes [18]. In the case that discrete values with probabilities are used, the scenarios to compute the impact are all the possible values with their associated probability.

With these scenarios, it is possible to compute the impact of the risk on the affected criteria. Then, the risk is evaluated as the variation present in the criteria due to the different inputs' uncertainties. This variation is the parameter that is incorporated into the CBA function as a cost.

### 2.2.3. Qualitative Risks Definition

The steps to consider qualitative risks in the decision-making process are exposed in Figure 3.

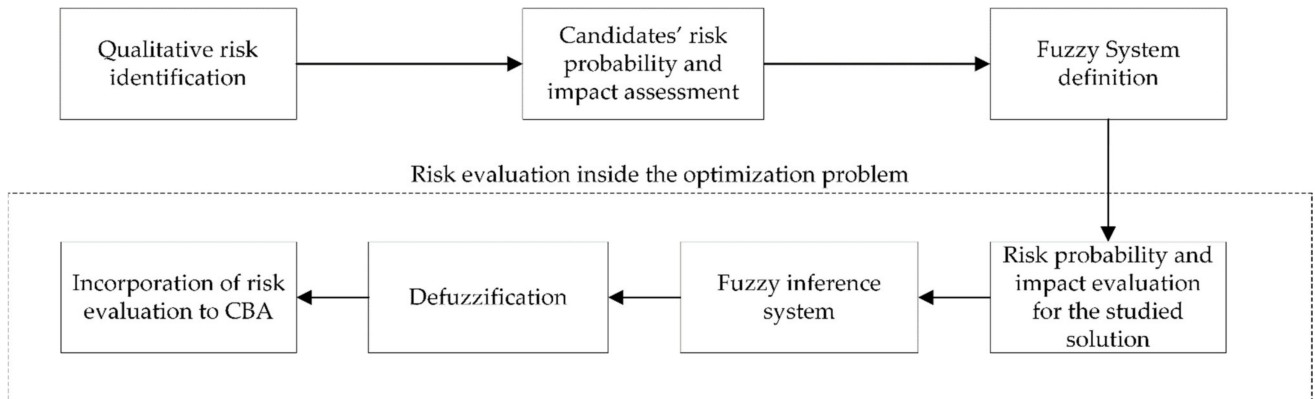

**Figure 3.** Qualitative risks treatment.

As commented previously, qualitative risks deal mainly with criteria that cannot be easily defined mathematically and that are approximated subjectively by decision-makers. This is the case of some social and environmental aspects which do not have clear measurement strategies, such as social welfare and local community perceptions. Once these risks are identified, it is required to evaluate and assign a numerical value to both their probability of occurrence and their impact on SME's objectives if they occur. Although there are other manners to define qualitative risks, the employment of probability and impact values, which is also suitable for quantitative risks, is the most appropriate one to deal with qualitative ones in decision-making problems in the industrial sector [28]. The assessment of probability and impact of qualitative parameters is done considering the decision-board experience in the sector, knowledge on local society obtained through interviews, government surveys, etc., and vary according to the equipment considered for installation and their size. In the proposed methodology, the optimisation of the energy investment RIDM problem is performed continuously evaluating all possible solutions, as a set of them that are pre-defined does not exist. Thus, it is required to implement a strategy for the specification of probability and impact of qualitative parameters based on the decision-maker's opinion for all possible solutions. This is done by the creation of a decision tree whose branches divide all the possible solutions in ranges of specific equipment and sizes at which the probability and impacts can be defined by decision-

makers. This decision tree is used by the continuous optimisation algorithm to identify the applicable impact and probability values for the solution that are iteratively analysed.

Although these probability and impact values can be defined as crisp values, they are unavoidably subject to judgemental vagueness. To avoid losing experts and decision-makers' valuable opinions, these parameters should not be considered as crisp, but as part of a continuous function. To do so, a set of fuzzy membership functions are defined, which serve as input for the Fuzzy Inference System (FIS) that computes the risk evaluation. Two FIS are widely accepted and employed in the literature; the Mamdani and the Takagi-Sugeno [29]. In this paper, the Mamdani method and the max–min inference are selected as they perform better in extracting experts' opinions on risk factors, and thus it is more suitable for RIDM problems [30]. In the Mamdani method, if–then rules and the implication method are used to obtain a fuzzy output, which has to be defuzzified in a later stage for its treatment in further mathematical equations. The if–then rules are designed to follow the logic of an expert risk assessor through a qualitative risk matrix [31], and the defuzzification is performed employing the centroid strategy, which provides solutions that naturally and smoothly respond to the created rules [32].

*2.3. Criteria Ranking*

In this stage, the criteria selected and the risks identified are ranked to reflect the preferences of industrial SME in the energy investment RIDM problem. To capture these preferences, an AHP is employed, which is a tool to methodologically determine the weights based on subjective preferences and which is suitable to incorporate various criteria of different nature [12], including non-risk, quantitative risks and qualitative risks. The AHP method decomposes the problem into a hierarchy, having the goal on top and structuring the criteria and risks into levels, as can be seen in Figure 4. In classic AHP applications, the set of studied alternative solutions are included in the hierarchy, and they are analysed in a bottom-up perspective, from sub-criteria to criteria preceding them in the hierarchy until reaching the overall goal. In this paper, as the evaluation of solutions is performed through a continuous optimisation problem, the AHP is employed to select the weights, which are later incorporated in the CBA function.

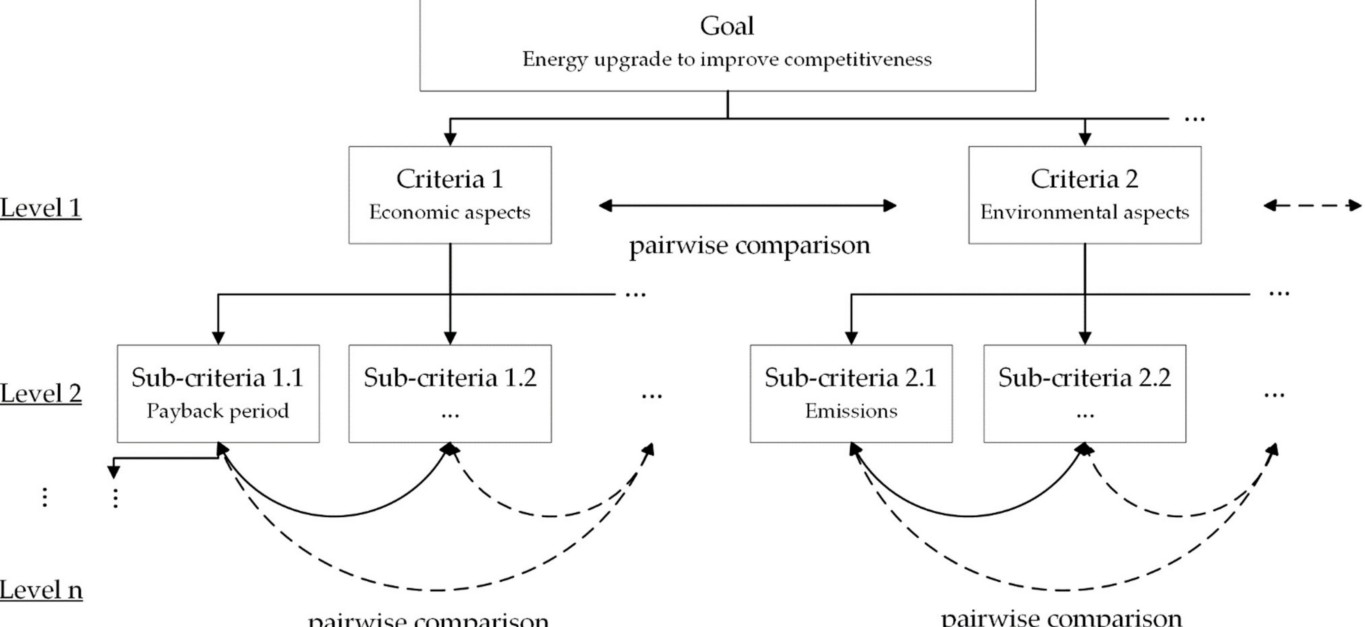

**Figure 4.** AHP hierarchy structure and pairwise comparison strategy.

The goal of the problem, located at the top of the hierarchy, is in this case the energy upgrade to become a prosumer and improve the competitiveness of the enterprise. Immediately below the goal, a set of criteria appears which designate the main aspects considered by the enterprise to reach the decision, such as economic and environmental aspects. Then, the next level details the criteria linked to these aspects and the relevant risks that apply. In this case, the sublevel below the economic criteria can be formed by the NPV, the payback period, and their variation according to uncertain inputs, whereas the environmental field can include $CO_2$ emissions or soil depletion. After generating the hierarchy, each of the items in a level is compared to the rest in the same level and under the same hierarchy branch in a pairwise manner [33]. This process is reflected in a paired comparison matrix, in which the element $a_{ij}$ denotes the importance of parameter $i$ in front of parameter $j$ following the Saaty scale definition [34], exposed in Table 1. This matrix definition process is done for the upper or lower diagonal part, being the parameter in the opposite part, $a_{ji}$, equal to $1/a_{ij}$. Thus, the resultant matrix has the following structure:

$$
\begin{bmatrix}
1 & 1/a_{12} & 1/a_{13} \\
a_{12} & 1 & 1/a_{23} \\
a_{13} & a_{23} & 1
\end{bmatrix}
\tag{1}
$$

Based on this matrix, the weights can be computed using the geometric mean and multiplying the results of the matrix from the lower levels of the hierarchy until reaching the goal [35].

**Table 1.** Saaty fundamental AHP scale.

| Intensity of Importance | Definition |
|---|---|
| 1 | $i$ and $j$ are equally important |
| 3 | $i$ is moderately more important than $j$ |
| 5 | $i$ is strongly more important than $j$ |
| 7 | $i$ is very strongly more important than $j$ |
| 9 | $i$ is extremely more important than $j$ |
| 2,4,6,8 | Intermediate values between two adjacent judgements employed when compromise is needed |

*2.4. Optimisation*

Once the criteria and risks have been identified and ranked, establishing the framework for selecting the best trade-off energy investment to upgrade the energy infrastructure of the plant, it is possible to enter the optimisation stage. In this stage, the possible energy infrastructures are evaluated iteratively to reach the optimal energy investment decision. This is done by incorporating all the criteria and risks into a CBA, which forms the optimisation's objective function. In the CBA, the parameters that are beneficial and want to be maximised are included as benefits, such as the NPV of the investment and the social acceptance of the solution. In contrast, the parameters that represent a disadvantage or hazard are introduced as costs. This is the case, for example, of emissions and NPV variability. As these factors present different units, their value is normalised for its inclusion in a single function. This normalisation process is performed both to remove the dimensions and also to balance possible magnitude differences that exist between different criteria [36]. The transforming approach employed here, which is considered one of the most robust regardless of the original range of parameters [37], is:

$$
p^{trans} = \frac{p - p^0}{p^{max} - p^0}
\tag{2}
$$

where $p^{trans}$ is the normalised parameter which lays between 0 and 1, $p$ is the measured value and $p^0$ and $p^{max}$ are the minimum and maximum values achievable, respectively.

Once the parameters are normalised, they are included in the CBA function with the weights obtained in the AHP.

Some of the parameters included in the CBA function are related to the performance of the energy infrastructure over time, and for this reason, it is required to compute the operation of the upgraded plant for the expected lifetime of the energy investment. This is performed by modelling the plant employing the Energy Hub (EH) concept [38], which can be expressed mathematically as:

$$L = \eta P \tag{3}$$

where $L$ represents the demand of the plant or power output, $P$ the generation or power input, and $\eta$ the connectivity matrix, which includes the dispatch factors and the efficiency of the equipment. This model represents the power balance which has to be fulfilled at all times which, together with other restrictions such as power exchange thresholds with external grids and equipment operation bounds, serves as the basis to evaluate the operation of the plant and obtains relevant parameters which should be included in the CBA such as the NPV and payback period.

The CBA obtained from the different criteria is optimised, aiming to reach as many benefits as possible with the least costs. To do so, a stochastic global algorithm is employed, which assures the surveillance of the entire search space and has better chances to find the global optimum compared to other optimisation methodologies [39]. In this paper, the Direct Search (DS) global optimizer is employed due to its capabilities to reach the global solution efficiently. Through this method, the search space surveillance is performed through the selection of a set of possible solutions or candidates, which are evaluated for the problem under study. The first set of candidates is computed based on an initial point provided by the decision-maker, which can be any point in the search space. The algorithm adds the unitary pattern vectors to the initial point, creating the first mesh. All the points in the mesh are possible energy-investment solutions whose CBA is evaluated. The results of these energy-investment possibilities enable the algorithm to move in the search space, creating new meshes having as starting points those in the previous mesh that provided favourable results, approaching the global optimum efficiently. The calculations required to compute the CBA and optimize it depend on the specific case study considered and the criteria and risks identified.

### 2.5. Methodology Generalisation

The exposed methodology has been designed for RIDM energy investment problems in industrial SMEs, addressing the challenges globally faced by these entities and creating a solid framework for the assessment of new energy equipment and management solutions. As can be inferred from previous paragraphs, this methodology can be divided into three different strata:

1. Input information from enterprise characteristics and the framework at which it operates.
2. Risks, factors and limitations applicable to the energy-investment problem.
3. Mathematical strategies, techniques and tools for the proper incorporation of factors different in nature in an optimisable function.

All three points are directly applicable to worldwide SMEs that face the energy-investment decision-making problem. However, and although the proposed methodology has been especially designed for these entities in the energy-investment context, it is expandable to other decision-making problems.

For instance, the current energy-investment problem faced nowadays by managers of buildings, communities or districts can also be assessed through the proposed methodology. In such cases, the methodology should be modified to incorporate as input information the specific data and characteristics of tertiary and residential sectors, such as:

- Space occupation and energy consumption demand at different conditions;
- Consumers' flexibility and load shifting behaviour;

- Compatibility of energy equipment with building/community/district purposes; and
- Integration with Smart City initiatives, etc.

The constraints that apply to the energy investment itself may also differ, focusing more on operational benefits and allowing larger payback periods. Moreover, as in these entities the human factor is much stronger than in the industry, issues related to social welfare, environment and safety should be considered as determinant criteria, having economic criteria either in the same level or moved to the background. Despite these differences with the industrial SMEs' problem treated in this paper, energy-investment problems deal with a similar mixture of criteria which have to be evaluated along the lifetime of the infrastructure. For this reason, the mathematical strategies, techniques and tools exposed for suitably address the energy-investment RIDM problem are applicable not only to industrial SMEs' problems, but also to other entities facing the challenge to perform an energy investment with minimum risks.

Furthermore, if the energy investment is not performed by SMEs or individuals but by governmental entities or big corporations, the proposed methodology can be adapted to incorporate the possibility to carry out multiple-phase investments and project expansions. In this case, the inputs of the system should incorporate the time frames at which investments are desired and the growing energy requirements to be fulfilled.

Apart from its application to energy-investment problems from a wide point of view, the proposed methodology can also expand to suit other RIDM problems not directly related to energy issues, but with other tangible assets, such as the placement and investment of distribution centres. For this case, the inputs should incorporate the expected products' traffic, location of stakeholders and clients, earth-moving constraints, etc. Additionally, for distribution and logistics centres, the investment problem is not only economic, and constraints are closely related to the acceptance of the local community since it can strongly affect the structure of the environment and the communications infrastructure of the district and area in which it is placed, due to important visual impact for the community. Therefore, and in a similar way to the case of energy-investment in non-industrial entities, it is possible to use the proposed methodology, strategies and tools to evaluate the selected criteria and the qualitative and quantitative risks that should be considered to make the decision.

Thus, it can be concluded that the proposed methodology can be applied to a vast number of decision-making problems in which quantitative and qualitative risks have to be evaluated. For these new applications, the general methodology and tools can be maintained while the inputs of the system should be modified to suit the specific problem to be addressed as well as the application constraints. In this way, it is possible to employ the proposed strategies and tools to reach the balanced trade-off solution that best reflects the interests of the entity making the decision.

## 3. Case Study

In this section, a case study for an industrial SME of the automotive sector is presented in which the methodology exposed in the previous section is applied. Industrial SMEs, in contrast with other entities in the tertiary and residential sector, have higher thermal consumption than electrical consumption [40–42] and are characterized by a diversity of processes and equipment that enable the incorporation of different energy assets to interconnect the different energy carriers present in the industry, increasing the robustness of the energy system [43]. Additionally, the load pattern of industrial SMEs is much more predictable than in other sectors, as it is strongly affected by production and varies only slightly with daily human behaviour [44]. This is especially true for the case of industrial SMEs of the automotive sector, as they do not have stocks and produce in a just-in-time manner to supply materials and components to other enterprises for continuous vehicle manufacturing [45,46], thus presenting a much more stable load curve.

The case study exposed in this section is based on a real industrial SME of the automotive sector and reflects the main characteristics exposed of overall industries and especially of those related to the automotive sector. The annual electrical and thermal demands of the

industrial plant are 679,240 MWh and 1,127,600 MWh, respectively; and an example of the demand pattern followed in one day can be seen in Figure 5.

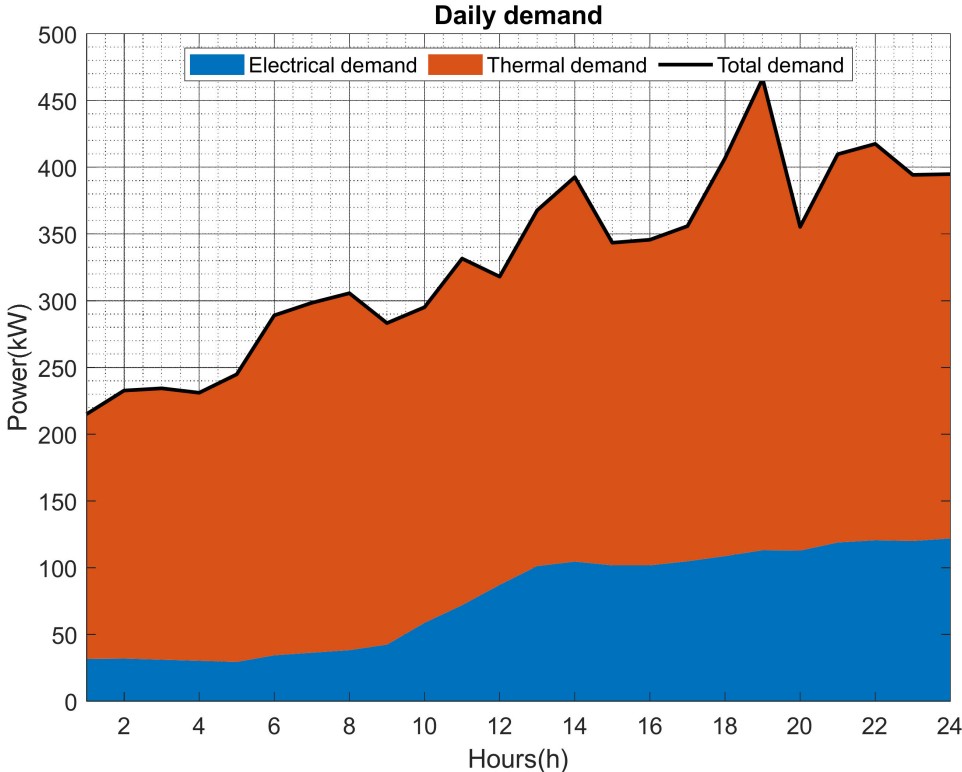

**Figure 5.** Daily load demand for the case study industrial SME.

In the following subsections, each of the stages of the proposed methodology are developed with the objective to achieve the best energy-investment decision in accordance with the objectives and characteristics of industrial SMEs.

### 3.1. Scope, Context and Criteria

The considered industrial manufacturing plant wants to upgrade its energy infrastructure to improve its competitiveness. This can be done by incorporating RES and other equipment to enhance its efficiency and reduce its carbon footprint, and to explore the capacity of exchanging electricity with the utility grid by adopting an active prosumer model.

Currently, the plant fulfils its electrical and thermal demands through the direct purchase of electricity and the combustion of natural gas in a boiler. The boiler equipment is foreseen to continue in operation for the next 15 years and thus its substitution is not evaluated. The enterprise has 12,000 m² of available space for the installation of a PV system, and it is also considering the inclusion of a Combined Heat and Power (CHP) unit, a Heat Pump (HP), Thermal Storage System (TSS) and an Electrochemical Storage System (ESS). However, the maximum investment is limited to 1,000,000€ and the payback period has to be lower than 6 years. With this context and scope, the combination of criteria proposed in this article to evaluate the best energy-investment decision is shown in Table 2.

**Table 2.** Criteria for the energy-investment decision-making problem case study.

| Criteria | Sub-Criteria | Description |
|---|---|---|
| Economy | NPV<br>Business continuity | Value of the investment at the end of its expected lifetime.<br>Investment influence on supporting business continuity in the future. |
| Technology | Innovation<br>Maturity<br>Safety | Competitive advantage through innovation.<br>Feasibility of the technological solutions to be integrated into the SME.<br>Safety of the solution for workers and local community. |
| Social | Social benefits<br>Social acceptance<br>Administration alignment | Contribution to the advancement of society.<br>Attitudes of users on the energy infrastructure upgrade.<br>Alignment of the solution with administrative and legislative energy trends. |
| Environment | Pollutant emissions<br>Ecology influence | Emissions of greenhouse gases to the atmosphere.<br>Direct and indirect influences on ecosystem. |

### 3.2. Risk Identification and Analysis

Keeping in mind the criteria selected, the identified quantitative and qualitative risks that affect them for this case study are exposed in Table 3. In the following pages, each of these risks is characterized for its inclusion in the optimisation problem.

**Table 3.** Risks identified for the energy-investment decision-making problem case study.

| Risk ID | Risk Description | Criteria Affected | Risk Type [1] |
|---|---|---|---|
| 1 | Electricity cost market uncertainty | NPV | QT |
| 2 | Gas cost market uncertainty | NPV | QT |
| 3 | Feed-in tariff uncertainty | NPV | QT |
| 4 | Emissions cost market uncertainty | NPV | QT |
| 5 | PV O&M [2] costs uncertainty | NPV | QT |
| 6 | Electrochemical storage O&M costs uncertainty | NPV | QT |
| 7 | Business continuity subjectivity | Business continuity | QL |
| 8 | Innovation subjectivity | Innovation | QL |
| 9 | Maturity subjectivity | Maturity | QL |
| 10 | Safety subjectivity | Safety | QL |
| 11 | Ecology influence subjectivity | Ecology influence | QL |
| 12 | Social benefit subjectivity | Social benefit | QL |
| 13 | Social acceptance subjectivity | Social acceptance | QL |
| 14 | Administrative alignment subjectivity | Administrative alignment | QL |

[1] QT = quantitative; QL = qualitative. [2] O&M= operation and maintenance.

#### 3.2.1. Quantitative Risk Analysis

Here, the quantitative risks are analysed and a numerical description assigned to them.

- Risks 1–4 These risks correspond to the uncertainty in the forecast of future market costs, including the price of electricity, gas and emissions as well as the feed-in tariff at which electricity is sold. The uncertainty of the increment ratio of energy and emissions costs creates different operation and financial scenarios for which the studied solutions provide distinct results on the criteria. These uncertainties and criteria variation have to be evaluated as a risk in the decision-making process. To do so, the future scenarios represented as price increments possibilities obtained from the literature are analysed. These scenarios, which present an equal probability of occurrence, are exposed in Table 4.

**Table 4.** Risks 1–4 numerical description.

| Risk ID | Factor Description | Scenarios | Source |
|:---:|:---:|:---:|:---:|
| 1 | Electricity cost yearly percentage increase | [1.40;4.06;4.82] | [47] |
| 2 | Gas cost yearly percentage increase | [0.65;1.40] | [48] |
| 3 | Percentage of the electricity cost at which electricity is sold | [0.80;0.9] | [49] |
| 4 | Emissions cost yearly percentage increase | [1.14;6.45] | [50] |

- Risks 5–6 These risks relate to the fact that PV and electrochemical energy storage systems are growing in adoption, decreasing their operation and maintenance (O&M) costs as a consequence of the economy of scale that the sectors are experiencing, although the cost evolution is not clear yet. To capture this uncertainty, the cost decrease expectation is extracted from the literature and the possible scenarios, also with equal probabilities and exposed in Table 5, are analysed under the point of view of its impact on the criteria for RIDM.

**Table 5.** Risks 5–6 numerical description.

| Risk ID | Factor Description | Scenarios | Source |
|:---:|:---:|:---:|:---:|
| 5 | PV O&M costs yearly percentage decrease | [0.5;0.95;1.7] | [51] |
| 6 | Electrochemical storage O&M costs yearly percentage decrease | [3.3;3.7;4.5] | [50,52] |

All these quantitative risks affect the NPV criteria. To evaluate this risk, the impact in the NPV is computed for all the risk scenarios combinations, obtaining, as a result, the variation of the NPV. This NPV variation is included in the CBA function aiming at its reduction for risk minimisation.

### 3.2.2. Qualitative Risk Analysis

Risks 7–14 are qualitative and thus they are defined based on the opinion of decision-makers and experts. To capture their knowledge, decision-makers perform an analysis of the probability of risks to happen and the impact these would have on the enterprise's objectives depending on the energy infrastructures evaluated. As the energy investment RIDM is optimised continuously, all possible energy infrastructure that could be a solution have to be assessed. To do this analysis, decision-makers rely on their experience and knowledge of the local community, legislation trends and company environmental and social commitment, as well as initial enterprise's constraints such as maximum investment. The probability and impact evaluations are reflected into decision trees allowing the optimisation algorithm to obtain these risks' values for the evaluated energy-investment solutions. As probability and impact are not necessarily distributed in the same ranges of equipment, for each studied risk one decision tree is required for probability and another for impact. Therefore, in the case study presented here, a total of 16 decision trees are constructed. The resultant decision trees for the decision-making problem are subjective, as they derive from the opinions of experts considering previous experience surveys performed to users and local social agents. An example of a decision tree is exposed in Figure 6. This decision tree serves to specify the impact of the solution on business continuity according to the equipment selected. A higher value means that the studied solution has a higher impact than other solutions, being a high impact desirable. In this case, the decision-makers specify that business continuity should not have a big CHP installation, whereas it is positive to include a PV system, although in a moderate manner. Of course, this assessment can change depending on the location of the company, the production sector, local trends and opinions about the industries, etc.

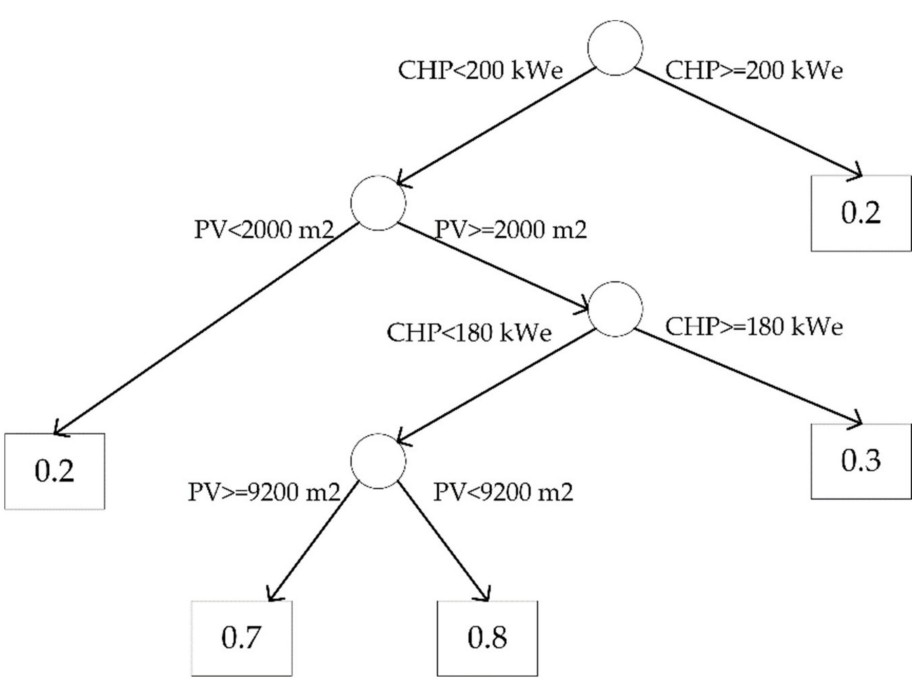

**Figure 6.** Decision tree for Risk 7 impact, business continuity.

The probability and impact values specified by decision-makers are influenced by vagueness, as one person can understand 0.7 to be a moderate impact, whereas another one can understand it to be high. To avoid losing information regarding the true meaning behind the value specified by the decision-maker, a fuzzy strategy is employed in which probability and impact can correspond to one or more fuzzy membership functions that serve to compute the risk evaluation through the FIS. In Table 6, the membership functions employed for probability, impact, and risk evaluation are exposed. In this case study, the employed membership functions specified in the last column of Table 6 are trapezoidal. Their definition is performed in the $(a_1, a_2, a_3, a_4)$ form, which correspond to the specific function's shape such that:

$$f(x; a_1, a_2, a_3, a_4) = \begin{cases} 0, \ (x < a_1) \ or \ (x > a_4) \\ \frac{x-a_1}{a_2-a_1}, \ a_1 \leq x \leq a_2 \\ 1, \ a_2 < x < a_3 \\ \frac{a_4-x}{a_4-a_3}, \ a_3 \leq x \leq a_4 \end{cases} \tag{4}$$

**Table 6.** Fuzzy membership functions and linguistic description of risk impact, probability and evaluation.

| Risk Aspect | Linguistic Definition | Fuzzy Number |
|---|---|---|
| Probability | High | (0.6, 0.9, 1, 1) |
| | Medium | (0.2, 0.4, 0.6, 0.8) |
| | Low | (0, 0, 0.1, 0.4) |
| Impact | Large | (0.7, 0.9, 1, 1) |
| | Considerable | (0.5, 0.7, 0.8, 0.9) |
| | Moderate | (0.2, 0.4, 0.6, 0.8) |
| | Minor | (0.1, 0.2, 0.3, 0.4) |
| | Negligible | (0, 0, 0.1, 0.2) |
| Evaluation | High | (0.6, 0.9, 1, 1) |
| | Medium | (0.2, 0.4, 0.6, 0.8) |
| | Low | (0, 0, 0.1, 0.4) |

The membership functions for risk impact, probability and evaluation can be seen graphically in Figure 7.

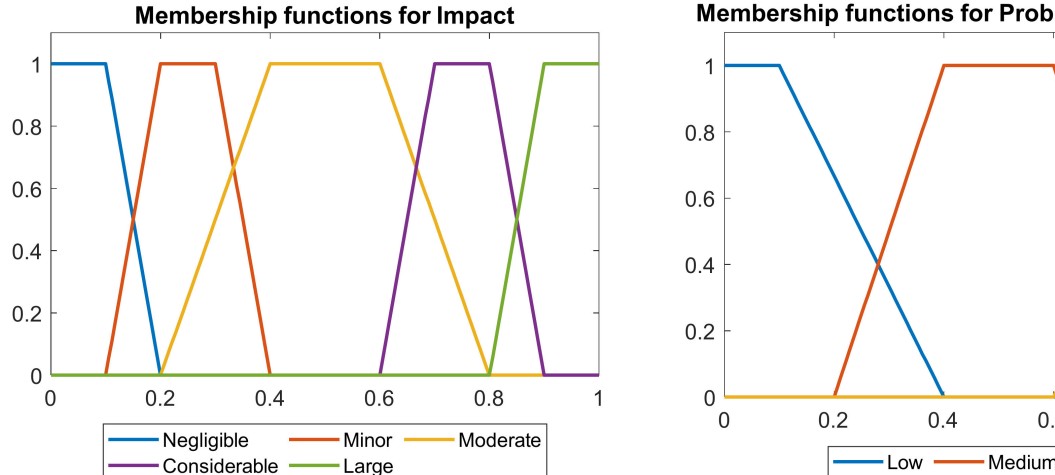

**Figure 7.** Risk impact, probability and evaluation membership functions.

Once the linguistic terms, fuzzy sets and decision trees for impact and probability assessment of candidate solutions are defined, the 15 required if–then rules for the Mamdani FIS are designed, which enable us to compute the risk evaluation that has to be included in the CBA function. To support their creation, the qualitative risk matrix shown in Table 7 is generated, where the risk evaluation fuzzy set is identified based on the risk probability and impact specified by the decision-maker.

**Table 7.** Qualitative risk matrix for the case study.

| | | Impact | | | | |
|---|---|---|---|---|---|---|
| | | **Large** | **Considerable** | **Moderate** | **Minor** | **Negligible** |
| **Probability** | **High** | High | High | High | Medium | Medium |
| | **Medium** | High | High | Medium | Low | Low |
| | **Low** | Medium | Medium | Low | Low | Low |

From this matrix, the rules for the Mamdani FIS are generated. As an example, five of them are shown here:

*If (Probability is High) and (Impact is High), then risk is High*
*If (Probability is High) and (Impact is Moderate), then risk is Moderate*
*If (Probability is Medium) and (Impact is Considerable), then risk is High*
*If (Probability is Medium) and (Impact is Moderate), then risk is Moderate*
*If (Probability is Low) and (Impact is Minor), then risk is Minor*

Here, an example of the working behaviour of the developed FIS is exposed to assess the business continuity when analysing the possibility of installing 6000 m$^2$ of PV, a CHP system of 180 kW$_e$ and an HP of 150 kW. According to the decision tree exposed previously, the impact of this solution on business continuity is 0.3. For the case of the probability of contributing to business continuity, the resultant value is 0.5, which has also been established following decision-makers' judgments. With this information, the risk can be evaluated through the FIS as exposed in Figure 8. According to the fuzzy membership functions used, the probability parameter belongs only to one membership function. In contrast, the impact value belongs to two membership functions as it can express either

a minor impact or a moderate impact. Thus, it is necessary to analyse two rules: one for medium probability and moderate impact and another for medium probability and minor impact. These two rules lead to two possible risk evaluations, which are combined to consider judgemental vagueness.

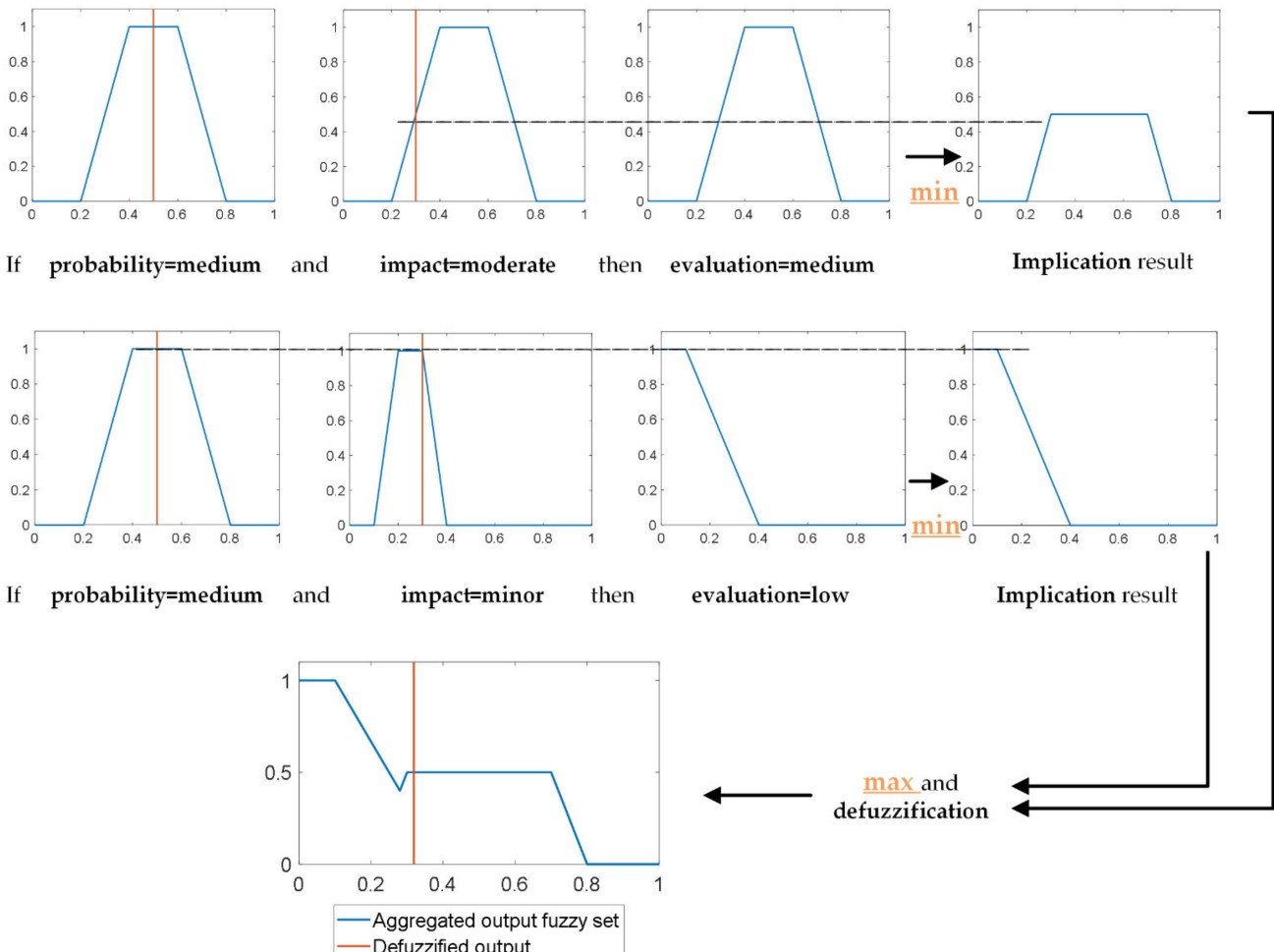

**Figure 8.** FIS procedure for evaluating the business continuity of a candidate solution for the case study developed.

In the first activated rule, the obtained risk is medium. As the value of the impact is 0.3, it belongs to the moderate impact membership function although it does not completely fulfil it. For this reason, the implication is performed through max–min composition to reduce the influence of this rule in the output according to the degree of fulfilment of input membership functions [53]. In the second activated rule, the obtained risk is low and the min operator is not activated, as both membership functions are completely fulfilled. These two outputs are aggregated, obtaining the fuzzy risk evaluation, which is defuzzified through the centroid method. The centroid returns the centre of gravity in the *x*-axis of the area under the membership function and is a consistent method suitable for one-dimensional output problems where no real-time implementation occurs, such as the one presented here [54]. In the example here shown, the defuzzification returns a final value of 0.3188, which is the measure of risk evaluation included in the CBA function.

### 3.3. AHP Ranking

Being the main goal of the energy upgrade of the company to improve the competitiveness, a hierarchy with all the criteria and risks identified is constructed, which can be seen in Figure 9. The first level is formed by the main decision criteria, which are economy, technology-based, social, and environmental criteria. Then, each of these criteria

are sub-divided into several items that are those already exposed in previous sections, specifically in Table 2. To these items, the NPV variation is included as a sub-criterion arising from the consideration of quantitative risks.

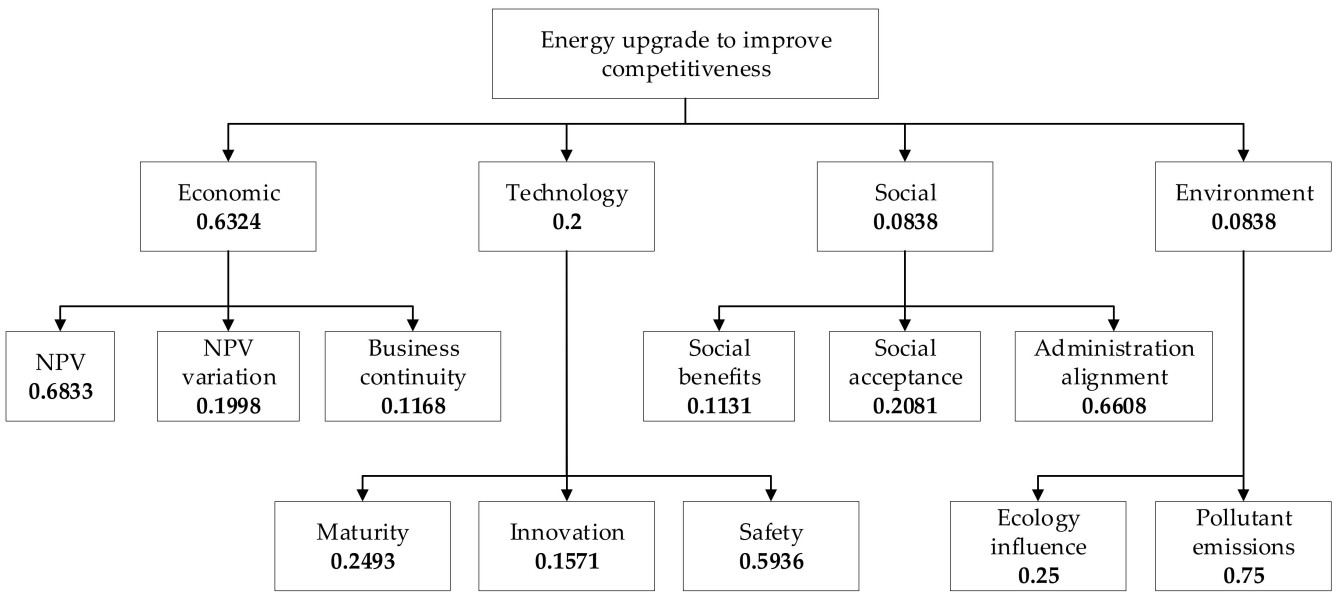

**Figure 9.** Criteria hierarchy with their corresponding level weights for the case study.

Once the structure is created, the criteria in the same level are compared in a pairwise manner using the Saaty fundamental scale, and the weights for this level are obtained. This process is performed by decision-makers considering the interests of the enterprise and the importance of each of the elements in terms of its predecessor in the hierarchy. These preferences are independent of the value that the criteria and risks take when evaluating possible energy infrastructure upgrades. Therefore, they are maintained constant, reflecting the preferences of the enterprise, and appear in the CBA function multiplying the value of criteria and risks, which change for every solution analysed, to assure a balanced trade-off suitable for the industrial SME.

As an example of the application of the Saaty fundamental scale, the comparison matrix and resultant weights for the first hierarchy level, where the main criteria are placed, are exposed in Table 8. The weights, as specified in Section 2.3, are obtained through the geometric mean, expressed as:

$$W_i = \left( \prod_{j=1}^{n} W_{ij} \right)^{\frac{1}{n}} \tag{5}$$

where $W_i$ is the obtained weight, $W_{ij}$ represent the comparison performed between parameters in row $i$ and column $j$ and $n$ is the total number of parameters in the same layer for comparison. Given that the Saaty scale and the geometric mean can produce weights greater than one, once all the weights in the same layer are obtained, these have to be normalised:

$$W_{i,norm} = \frac{W_i}{\sum_{i=1}^{n} W_i} \tag{6}$$

**Table 8.** Pairwise comparison matrix for the second hierarchy level.

|  | Economic | Technology | Social | Environment | Weights |
|---|---|---|---|---|---|
| Economic | 1 | 5 | 6 | 6 | 0.6324 |
| Technology | 1/5 | 1 | 3 | 3 | 0.2 |
| Social | 1/6 | 1/3 | 1 | 1 | 0.0838 |
| Environment | 1/6 | 1/3 | 1 | 1 | 0.0838 |

This process is repeated for all the sub-criteria, obtaining the weight hierarchy structure shown in Figure 9.

With this information, the global weights of the sub-criteria for their incorporation in the CBA function are computed through the multiplication of the resultant weights in a bottom-up perspective:

$$W_{i,\ global} = W_{i,normL2} \times W_{predecessor,norm} \qquad (7)$$

where $W_{i,global}$ represents the global weight of a parameter, $W_{i,normL2}$ the normalised weight obtained for the parameter in the second layer of the diagram through pairwise comparison, and $W_{predecessor,norm}$ the normalised weight of its predecessor in the hierarchy.

The global weights for all the considered criteria and risks are exposed in Table 9, together with the symbols employed to express them in upcoming mathematical equations.

**Table 9.** Global weights for the sub-criteria in the analysed case study.

| Sub-Criteria | Symbol | Weight |
|---|---|---|
| NPV | $NPV$ | 0.4321 |
| NPV variation | $NPV_V$ | 0.1264 |
| Business continuity | $BC$ | 0.0739 |
| Maturity | $M$ | 0.0499 |
| Innovation | $IN$ | 0.0314 |
| Safety | $SF$ | 0.1187 |
| Social benefits | $SB$ | 0.0110 |
| Social acceptance | $SA$ | 0.0174 |
| Administration alignment | $AA$ | 0.0554 |
| Ecology influence | $EI$ | 0.0629 |
| Pollutant emissions | $PE$ | 0.0210 |

*3.4. Optimal Energy-Investment Selection Process: Continuous CBA*

Once the criteria are selected and ranked, it is possible to proceed to the optimisation of the energy-investment RIDM for the industrial SME. The variables to optimise are the equipment to install and their sizes, whereas the constraints include the maximum investment that can be performed by the enterprise and the maximum allowable payback period. The objective function of the optimisation problem is the CBA function, where all the criteria and risks are considered either as a benefit or as a cost, including the quantitative and qualitative risks. This CBA function is maximised, aiming for an energy infrastructure that creates as many benefits as possible with low costs. The benefit criteria are those attributes included as positive terms and which wish to be maximised, while the cost criteria are those included as negative terms and that want to be kept as low as possible. For the present case study, bearing in mind the weights obtained through AHP, the resultant CBA function is:

$$\begin{aligned} f = 0.4321NPV &-0.1265NPV_V + 0.0739BC + 0.0499M + 0.0314IN \\ &+0.1187SF + 0.011SB + 0.174SA + 0.0554AA - 0.0629EI \\ &-0.0210PE \end{aligned} \qquad (8)$$

It can be seen that almost all criteria are incorporated with a positive value, being the NPV variation, ecology influence and pollutant emissions, and the negative criteria

which represent a cost that have to be kept low. This CBA function has to be evaluated for all possible energy-investment solutions to upgrade the energy infrastructure of the plant, examined through the DS optimisation algorithm. The value of some of the criteria can be obtained directly from the selection of the energy infrastructure, the decision trees and the FIS exposed in previous sections. However, the NPV, NPV variation and pollutant emissions criteria require the computation of the infrastructure operation along the lifetime of the equipment and, in the case of the NPV, a comparison with the hypothetical situation of not performing any investment. For this reason, an analysis of the plant performance for the lifetime of the new equipment, which is considered to last for 15 years, is included inside the optimisation process. This analysis is carried out by employing the EH concept. For the studied industrial plant, the EH equilibriums for the electrical and thermal sides are stated as:

$$P_{PV}\eta_{PV} + P_{UG}\eta_{UG} + P_{CHP} + P_{DES}\eta_{DES} = \frac{P_{ED}}{\eta_{ED}} + P_{UGS} + \frac{P_{CES}}{\eta_{CES}} + P_{ET} \tag{9}$$

$$Q_{CHP} + Q_{BOI} + Q_{DTS}\eta_{DTS} + Q_{ET} = \frac{Q_{TL}}{\eta_{TL}} + \frac{Q_{CTS}}{\eta_{CTS}} \tag{10}$$

where $P_{PV}$, $P_{UG}$, $P_{CHP}$ and $P_{DES}$ are the electrical power coming from the PV system, the utility grid, the CHP system and the energy storage, respectively; $P_{ED}$, $P_{UGS}$, $P_{CES}$ and $P_{ET}$ are the electrical power to the internal demand, the one injected back to the utility grid, the employed to charge the energy storage and the sent to the HP system, respectively; and $\eta_{PV}$, $\eta_{UG}$, $\eta_{ED}$, $\eta_{DES}$, and $\eta_{CES}$ are the connectivity efficiencies with the PV system, the utility grid, the electrical demand and also the discharge and charge efficiencies of the energy storage. On the thermal side, $Q_{CHP}$, $Q_{BOI}$, $Q_{DTS}$ and $Q_{ET}$ are the thermal power generated by the CHP and the boiler and coming from the thermal storage and the HP; $Q_{TL}$ and $Q_{CTS}$ are the thermal power for thermal load and the one injected in the thermal storage; and $\eta_{TL}$, $\eta_{DTS}$ and $\eta_{CTS}$ are the connectivity efficiencies with the load and the discharge and charge efficiencies of the thermal storage.

These equilibrium equations are accompanied by restrictions that allow the EH to operate following the physical constraints existent in the real plant. These restrictions include equipment connectivity, power equipment operation bounds and external grid requirements. This mathematical model can be employed for the different energy infrastructures analysed and also for studying the operation of the current industrial plant, as it is possible to set equipment to any size including zero, maintaining the operationality of the infrastructure. With this model, the operation of the upgraded plant can be obtained through optimising its behaviour aiming at minimising costs, which serves for the computation of parameters that have to be included in the CBA function for assessing the suitability of the analysed energy infrastructure.

Considering these aspects and the methodology exposed in Figure 1, the energy-investment RIDM optimisation flowchart is detailed for this specific case study in Figure 10. First of all, the industrial plant, market information, uncertainty scenarios and decision-makers judgements data are obtained. This information is employed, in part, to compute the scenarios at which the performance and operation of the industrial plant are analysed for the non-risk criteria and quantitative risk criteria. After obtaining the scenarios data, the operation of the reference plant is computed, which reflects the situation if no energy investment is performed and the currently existing energy infrastructure continues in operation for the next 15 years. This reference plant computation serves as a base for comparison and calculation of the NPV for the analysed energy investments.

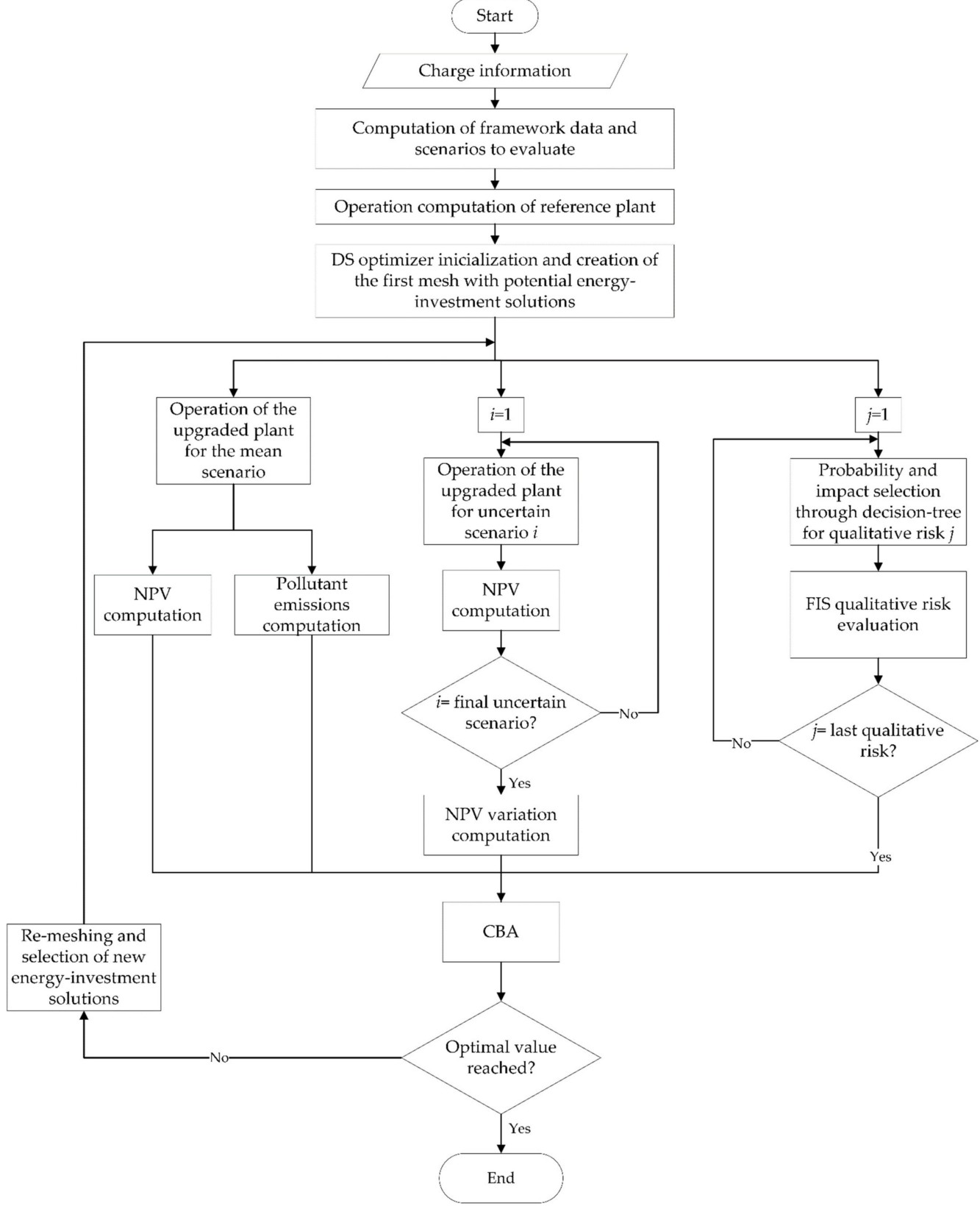

**Figure 10.** Optimisation flow chart for the case study.

Once this first part of the process is performed, the optimisation starts, aiming to find the energy investment that provides the best trade-off solution considering the risks related to its selection. Employing the DS algorithm, the first mesh is created through the

addition of pattern vectors to the initial point provided by the decision-maker. Each of the points in the mesh represents a candidate energy investment solution with a linked upgraded energy infrastructure, which is analysed for the non-risk criteria, quantitative risks, and qualitative risks. For the non-risk criteria, which are the NPV and pollutant emissions, the operation of the plant is computed for the whole expected lifetime. In the case of the quantitative risk, which is the variation of the NPV, the operation of the plant analysis process is repeated for all the considered uncertain scenarios. Then, for qualitative risks, the impact and probability are obtained through the decision trees, and the risk evaluation is computed by employing the designed FIS. The evaluation of all the criteria is included inside the CBA function, obtaining the expected benefits and costs and the suitability of the analysed solution. At this stage, the DS optimisation checks its finalisation constraints, which include, among others, the change tolerance in the CBA function and the achievement of a minimum step variation. If the algorithm has reached an optimal value, the process ends, obtaining the best energy investment for the enterprise and the upgraded energy infrastructure. If not, a new set of candidate solutions is generated by re-meshing the search space, considering the results of the last set of candidate solutions to approach the global optimal.

## 4. Results and Discussion

The results of performing the energy investment RIDM optimisation in the studied SME to upgrade its energy infrastructure are presented in this section. In order to evaluate the benefits of incorporating the risks into the decision-making problem, an optimisation considering only the non-risk criteria, which are the NPV and the emissions, has also been carried out. In Table 10, the initial investment and payback periods for both solutions, with and without risks, are exposed. Figure 11 depicts their NPV during the first 6 years, showing graphically the evolution of the investment and its return along time until the payback is achieved. The equipment selected by the optimiser for each of the alternatives is exposed in Table 11. As one of the investment solutions has been obtained through a without risks analysis whereas the other is the result of an optimisation accounting also with quantitative and qualitative risk factors, the energy infrastructure resulting from the different optimisation problems also present different consequences in terms of risk implications, which can vary the real outcome for the enterprise. To appreciate these implications, Table 12 has been created in which it is possible to see the value of all the criteria including risks for both optimisations. It is worth noting that for the without risks optimisation, risks have not been considered during the optimisation, but are computed at the end of the process for the sake of comparability.

**Table 10.** Energy investment main characteristics.

|  | **With Risks** | **Without Risks** |
| --- | --- | --- |
| Initial investment | 689,600€ | 909,960€ |
| Payback period | 3.4 years | 4.1 years |

**Table 11.** Energy equipment selected and their sizes.

| Equipment | Size | |
| --- | --- | --- |
|  | **Optimisation with Risks** | **Optimisation without Risks** |
| PV energy source | 12,000 m$^2$ | 12,000 m$^2$ |
| Thermal storage | 140 kWh | 135 kWh |
| CHP system | 140 kW$_e$ | 200 kW$_e$ |

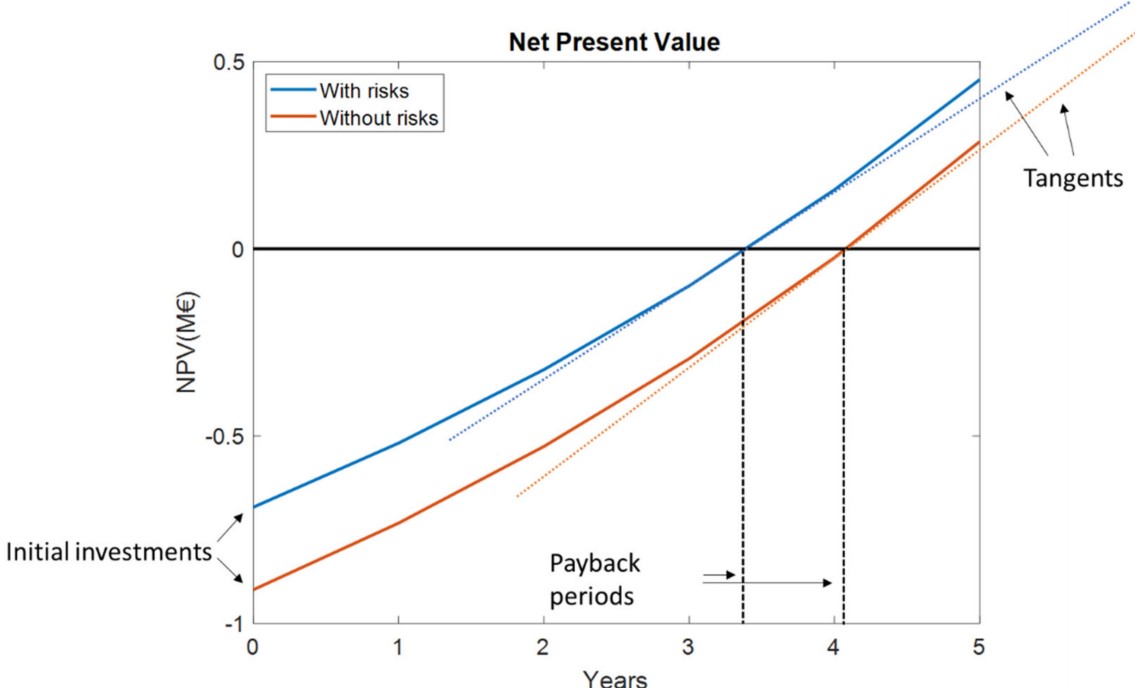

**Figure 11.** NPV evolution during the first 6 years after investment.

**Table 12.** Criteria evaluation for the two obtained solution.

| Criteria Evaluation | Value | |
|---|---|---|
| | Risks Included in the Optimisation Problem | Risks not Included in the Optimisation Problem |
| **Non-risk criteria** | | |
| NPV | 7,162,700 € | 7,470,000 € |
| Pollutant emissions (last year) | 306.136 tCO$_{2eq}$ | 307.4459 tCO$_{2eq}$ |
| **Quantitative risk criteria** | | |
| NPV range | 6,356,650–7,968,750 € | 6,616,350–8,323,650 € |
| **Qualitative risk criteria** | | |
| Business continuity | 0.8470 | 0.3188 |
| Maturity | 0.4071 | 0.5000 |
| Innovation | 0.8470 | 0.5902 |
| Safety | 0.5929 | 0.4071 |
| Social benefits | 0.1372 | 0.1372 |
| Social acceptance | 0.8470 | 0.8470 |
| Administration alignment | 0.8470 | 0.5000 |
| Ecology influence | 0.6263 | 0.6263 |

First of all, it is possible to see that, for the without risks case, the required energy investment is higher and the payback period is larger. It should be pointed out that these parameters are considered as constraints in the optimisation problem, specified as maximum allowable values chosen by the enterprise, but are not optimised. Instead, the objective for the without risks case is mainly the NPV maximisation while, for the case with risk, the objective is the trade-off between NPV, emissions and quantitative and qualitative risks. Therefore, the NPV for the without risk case analysis results higher than for the with risk one, as almost no other parameter is optimised. In Figure 11, it appears that although both initial investment and payback period are higher for the without risks case, its NPV line ascends at a higher grade, as exposed by the tangents of the graphs, obtaining more benefits per year and eventually surpassing the NPV for the case with risks. Although the

final economic value is more favourable for the without risks case, this solution does not consider any risk and creates an illusion of the investment's real profitability. Additionally, the NPV range, as exposed in Table 12, is higher for the without risks case, which reflects a less robust result where the final economic value is more uncertain and spans in a wider range which does, indeed, cover the NPV obtained for the with risks case.

Regarding the equipment selected, in both alternatives it is chosen to cover completely all the available area for the installation of the PV system together with thermal storage and a CHP system. The PV system is always chosen at its maximum capacity due to its low costs and, when considering risks, its positive influence in most of the evaluated qualitative criteria. In contrast, electrochemical storage and heat pump, which were also considered during the optimisation, do not appear as part of the new energy infrastructure. This is a consequence of the relationship between the economic benefits obtained from the equipment and their costs for the resultant energy infrastructure, which is not high enough to justify their incorporation. Additionally, when evaluating the risks, the influence of these equipment on the favourable risk criteria is not enough to include them regardless of their economic disadvantage. Despite these similarities between both solutions, when optimising the energy investment without considering risks, the size of the CHP system is significantly higher, which is the cause of the higher initial investment and larger payback period discussed previously.

Although the financial considerations exposed regarding the differences between the cases with and without risks are of importance for the SME, they only reflect a part of the global situation. In general, taking a decision considering only the non-risk criteria can lead to a situation with high exposure to strictly non-economic risks with great impacts on the enterprise. In this specific case study, not considering the risks leads to a solution that also compromises the qualitative risk criteria, having lower business continuity, safety and administration alignment, among others, as exposed in Table 12. For example, the solution obtained considering risks inside the optimisation decision-making problem evaluates that the contribution of the energy infrastructure to business continuity is 84.70%. In contrast, if this factor is not considered as criteria, as happens in the without risks optimisation, the contribution of the resultant infrastructure to business continuity is only 31.88%, reflecting the possibility of not supporting the company in future challenges. This variation in some of the qualitative criteria in the evaluated case study is a consequence of the danger related to CHP operation and the fact that these systems have been lately a focus of interest by governments, reducing the maximum installed capacity to reach a sustainable energy system and thus inhibiting further investments on them [55].

Thus, incorporating risk analysis in the energy-investment RIDM process enables the achievement of a solution that represents a trade-off between the considered criteria, allowing a smarter initial investment.

The energy investment obtained from considering all the risk and non-risk criteria enable the SME to upgrade its energy infrastructure and start acting as prosumer and, through the risk analysis performed, the operation of this energy infrastructure presents high reliability and robustness that supports the achievement of enterprise's objectives. For the case study analysed in this paper, the operation of the energy equipment and the exchange of energy with the utility grid are exposed in Figures 12–14. In Figure 12, it is possible to appreciate that electricity is being purchased when energy from the PV is not available, although at a smaller quantity than required by the internal demand. This is because part of this demand is fulfilled by electricity generated in the CHP system, which is employed both by the electrical side shown in Figure 12 and by the thermal one, shown in Figure 13. The energy exchange behaviour with the utility grid can be seen in Figure 14, where the electricity exchange with the utility grid is exposed together with the price of electricity in the wholesale market. It is possible to appreciate that, when the PV system is generating energy, this is employed for internal demand or to sell to the utility grid if the feed-in price is high enough and economic profit can be obtained. At the moments where electricity is sold, internal electrical demand is fulfilled by both the energy from the PV

not injected into the utility grid and the electricity coming from the CHP system. To adopt this optimal working behaviour, it is required to have a great synchronization between the electrical and thermal sides of the industrial plant. For this reason, it is beneficial to include thermal storage to support the mismatches between electrical and thermal demand and allow an optimal operation energy flow.

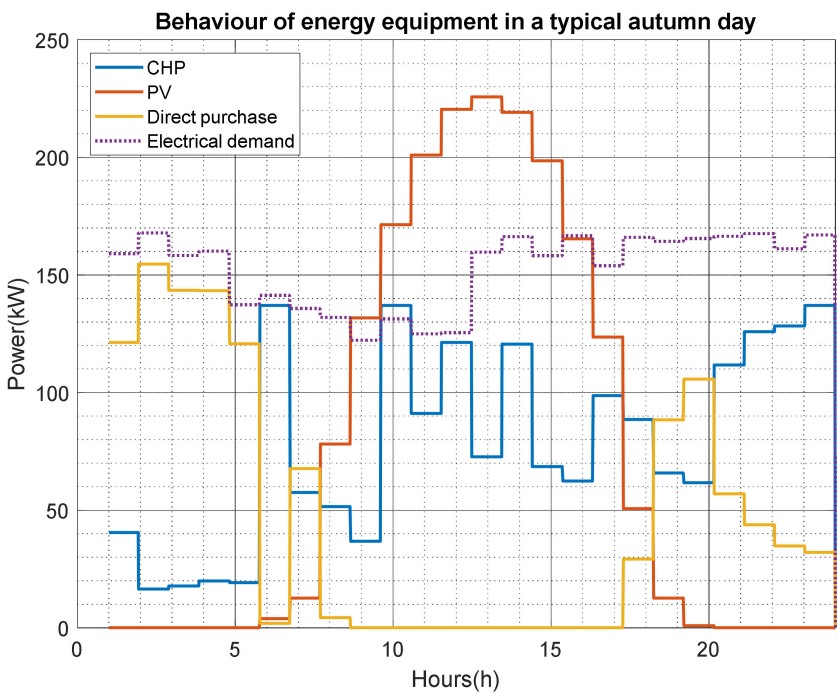

**Figure 12.** Electrical side energy equipment behaviour for the optimal energy investment.

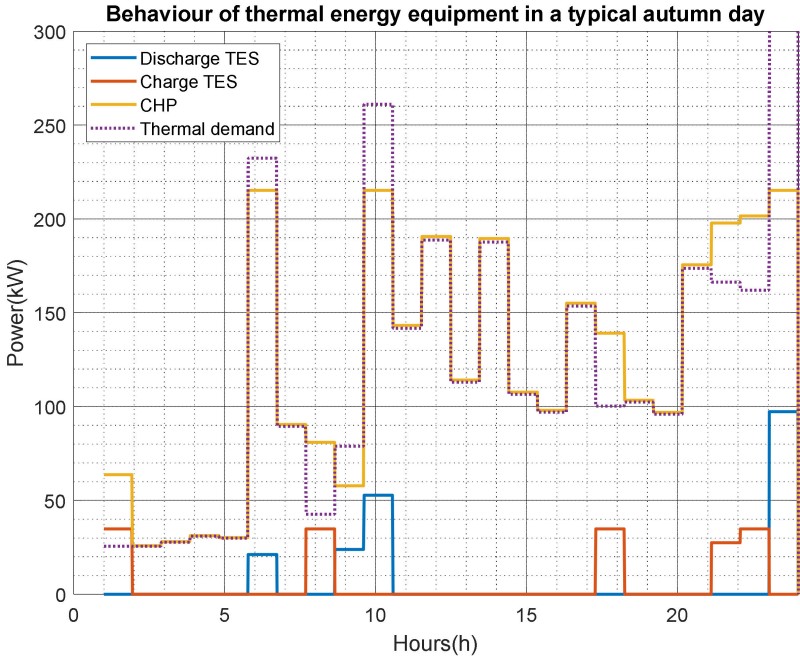

**Figure 13.** Thermal side energy equipment behaviour for the optimal energy investment.

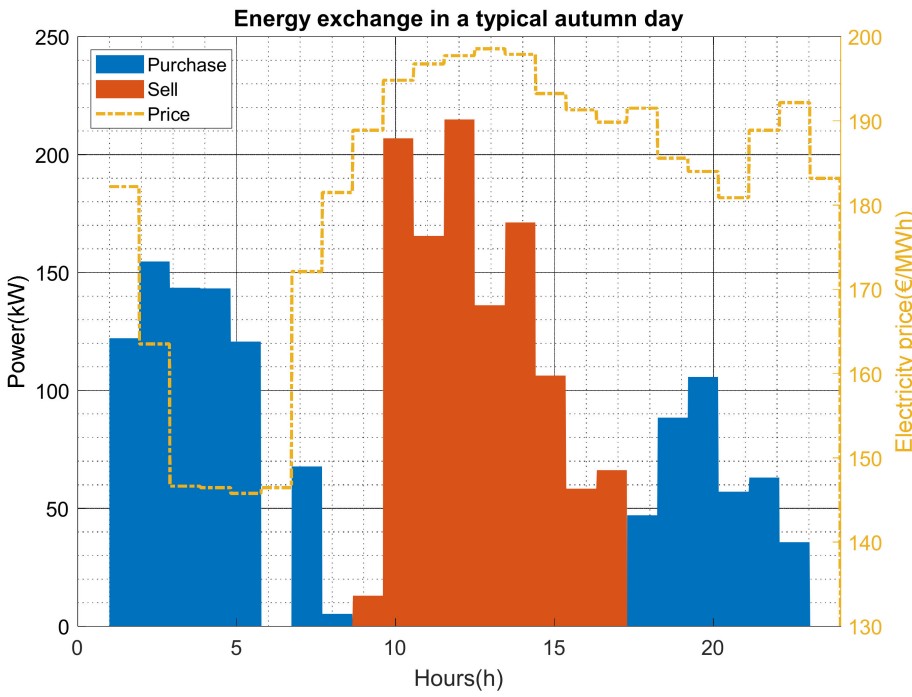

**Figure 14.** Energy exchange with the utility grid for the optimal energy investment.

## 5. Conclusions

This paper addresses the energy-investment optimisation problem to upgrade the energy infrastructure of an industrial SME to improve its competitiveness and support the green transformation by adopting an active role in the energy market. This energy investment optimisation problem is discussed considering all the relevant risks associated with the investment decision. A new methodology is proposed which incorporates the specification of the relevant criteria that apply for the industrial SME and the identification and characterization of quantitative and qualitative risks related to them. All these parameters are included in a single function through fuzzy logic and AHP weighting, performed with the support of experts and decision-makers. To reach the best-balanced trade-off solution for the SME, this function is optimized through Direct Search, a global optimisation algorithm that enables the surveillance of the continuous solution's search space. The created methodology, although especially designed to fulfil the requirements of industrial SMEs in upgrading their energy infrastructure, is expandable to other energy investment RIDM problems and also to problems related to the investment in other tangible assets. In these problems, decisions should also be taken considering a mixture of criteria including quantitative and qualitative measures of economic, technical, social, and environmental parameters along the expected lifetime of the investment. The weights granted to the different criteria in the decision-making process depend on the specific problem and its influences in the surroundings, which have to be specified by decision-makers. For this reason, it is required for decision-makers to have a deep knowledge on the interests of the entity taking the decision as well as on social, technical, political, and environmental local framework.

As a demonstration case, in this paper, the developed framework is applied to optimise the energy investment of an industrial SME based in a real manufacturing plant with the possibility to include a PV system, electric and thermal storage systems, a CHP system and an HP. Results show that employing a RIDM approach affects the optimal investment solution, reaching an energy infrastructure that represents a trade-off between the evaluated non-risk and risk criteria. Additionally, it is demonstrated that without incorporating the risk in the problem, industries would have to face the decision with incomplete information, reaching solutions that could be less beneficial and affect the future of the enterprise and trigger consequences on the surrounding community and environment. This conclusion

can be transposed to other entities performing investment decisions, as the omission of risks in the decision-making problem leads to solutions that do not consider possible impacts on the future, such as environmental effects or social welfare.

Thus, the methodology exposed in this paper presents a large practical value for both industrial SMEs and other entities where decision-making problems have to be addressed evaluating both quantitative and qualitative risks, as it can be modified and tailored to suit the specific problem addressed and its application constraints. This methodology can be adopted by decision-boards to analyse energy-investment problems and investment decisions on other tangible assets, enhancing the incorporation of criteria characterized by different nature in a single optimisation function and adjusting the input parameters to decision-makers requirements.

**Author Contributions:** Conceptualization, E.M.U., K.K. and L.R.; funding acquisition, E.M.U. and L.R.; methodology, E.M.U.; resources, E.M.U.; software, E.M.U. and V.M.-V.; supervision, K.K. and L.R.; validation, E.M.U.; visualization, E.M.U. and V.M.-V.; writing—original draft, E.M.U.; writing—review and editing, V.M.-V., K.K. and L.R. All authors have read and agreed to the published version of the manuscript.

**Funding:** This research has been funded by European Social Fund, the Secretariat of Universities and Research of Catalonia and the Generalitat the Catalunya under the grant number 2017 SGR 967.

**Institutional Review Board Statement:** Not applicable.

**Informed Consent Statement:** Not applicable.

**Data Availability Statement:** Not applicable.

**Conflicts of Interest:** The authors declare no conflict of interest.

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
