# Peer review of "Energy-Investment Decision-Making for Industry: Quantitative and Qualitative Risks Integrated Analysis"

_sustainability, doi:10.3390/su13126977_

Round 1
Reviewer 1 Report
-
The research objective and research questions are missing.
-
Please explain the limitations of the proposed method due to specific energy investments in different countries and regions.
-
Please better explain what motivated your work and why the proposed methodology should be published internationally. Why the case study has universal value?
-
Do not present results limited to your case study, but present a deeper synthesis of your results and present new theoretical insights.
-
Make sure that each abbreviation is explained at its first occurrence.
-
Please explain all graphical elements in more details.
-
Please state only new conclusions (that arise only from this work).
-
Please state only conclusions that are applicable to other fields and important to science.
Author Response
Dear reviewer,
We would like to thank you for all of your comments and suggestions, which have helped us to improve the quality of our manuscript and the presentation of the research done.
We have addressed them all, upgrading the manuscript as well as providing a specific response for each of the comments, which can be seen below.
Particularly, the research line of our paper has been highlighted, improving the exposition of the problem treated and including suitable references to describe the background of this scientific work. With this in mind, the objective of the paper has been stated more clearly as a follow-up of the published developments in energy investments and risk treatment in the decision-making process. The objective of our paper is the creation of a framework to properly address risk-informed decision-making for energy investment problems in industrial SMEs including both quantitative and qualitative risks, which fills a knowledge gap existent in the literature. The proposed methodology to address this problem and its implications for both industrial SMEs and other entities worldwide have been further discussed and the results and conclusions have been improved to better reflect the contributions of our manuscript.
In summary, we believe that the research problem, objectives, questions, methodology, tools, theoretical insights, results and discussion for both the specific case study and general applications have been exposed and discussed with greater rightness.
We believe the reviewed version is now suitable for publication in Sustainability.
Yours faithfully,
Eva M. Urbano

Reviewer 2 Report
The comments are in the attached file

Author Response
Dear reviewer,
We would like to thank you for all of your comments and suggestions, which have helped us to improve the quality of our manuscript and the presentation of the research done. Specially, we are grateful for the deep review performed, analysing the contents as well as the grammar and the ease of comprehension of the text over the full manuscript.
We have addressed all of them, upgrading the manuscript as well as providing a specific response for each of the comments, which can be seen in the attached file.
Particularly, the Introduction section has been modified to better introduce the topic of the paper and highlight the existent scientific background and research line in the literature that lead to the research objective stated in our study, which is the creation of a framework to properly address risk-informed decision-making for energy investment problems in industrial SMEs. Also, the Methodology and Case study sections have been improved providing more details on the techniques employed and their suitability for the problem of industrial SMEs, as well as the implications of our proposal for both industrial SMEs and other entities worldwide.
We believe the reviewed version is now suitable for publication in Sustainability.
Yours faithfully,
Eva M. Urbano

Reviewer 3 Report
This manuscript studies the energy investment decision-making for industries. Please see my comments below:
- Section 1 is well written. I am concerned with the novelties at the end of this section as I would expect all the contributions claimed have to be strongly supported by the methodology and evidence in the following sections.
- Section 2 is problematic to me. The framework presented is too generic and it can be applied for any types of assets. The explanations for those trivial flowcharts are lengthy, but energy-investment related issues are rarely explored in this section. What are the key contributions here? I don’t think it is enough to simply combine qualitative and quantitative and claim that as a novelty.
- Section 3 is a key section of this paper and I believe some parts of this section should be tailored and generalized to enrich the methodology.
- In Tables 2 and 3, please clearly show whether these are newly proposed or cite the sources for these criteria and risks.
- Please explain the rationale of fuzzy numbers in Table 6. What are “15 if-then rules” and the meaning of them?
- Risk IDs #1-7 have been explained; how about risk ID #8-14?
- Section 3.3: Please show the formulation or estimation of weight factors in Tables 8-1 & 8-2, and Figure 6? How can we add different terms representing economic, technology, social and environmental (I suppose they have different units)?
- Section 3.4: What type of optimization problem is this? What are the decision variables and constraints?
- Part of figure 7 is not visible.
- In general, there is some interesting information presented around the case study. However, it is poorly presented as academic research. I would recommend to focus on exploring and justifying the contributions claimed at the end of section 1. However, there need to have strong technical details on the modeling and optimization added for it to be considered for publication.
Author Response
Dear reviewer,
We would like to thank you for all of your comments and suggestions, which have helped us to improve the quality of our manuscript and the presentation of the research done.
We have addressed them all, upgrading the manuscript as well as providing a specific response for each of the comments, which can be seen in the attached file.
Particularly, the Introduction section has been modified to better define the contributions of the study, highlighting the existent scientific background and research line in the literature that lead to our research. The Methodology section has also been improved. More details have been provided on the requirements that the methodology has to fulfil for its application to energy investment risk-informed decision-making problem in industrial SMEs, creating a framework suitable for the inclusion of both quantitative and qualitative risks in a unique function which is assessed continuously considering the operation of the energy infrastructure over the lifetime of the equipment. The generalisation of this methodology to other energy-investment decision-making process has been discussed as well as its expandability to investment problems dealing with other types of tangible assets.
To better reflect the procedure specified by the methodology, both methodology and case study sections have been enriched with further explanations and mathematical formulation related to general techniques and tools employed and to the specific industrial SME modelling. Also, the contributions claimed at the beginning of the manuscript are assessed both in the methodology section and in the case study and are addressed in the conclusions section to reiterate the extension of the proposed solution in the paper to other applications.
In summary, we believe that the research problem, objectives, methodology, results and discussion for both the case study and general applications have been exposed and discussed with greater rightness, improving the presentation of the academic research done.
We believe the reviewed version is now suitable for publication in Sustainability.
Yours faithfully,
Eva M. Urbano

Round 2
Reviewer 1 Report
The purpose of the article could be better explained. The results are not sufficiently stated theoretically.
Reviewer 2 Report
Dear Authors,
thank you for all the answears. I do not have other remarks. I believe the paper is ready to be published.